# The Interplay of Autoclaving with Oxalate as Pretreatment Technique in the View of Bioethanol Production Based on Corn Stover

**DOI:** 10.3390/polym13213762

**Published:** 2021-10-30

**Authors:** Ali Hamdy, Sara Abd Elhafez, Hesham Hamad, Rehab Ali

**Affiliations:** 1Environmental Biotechnology Department, Genetic Engineering and Biotechnology Research Institute (GEBRI), City of Scientific Research and Technology Application (SRTA-City), New Borg El-Arab City P.O. Box 21934, Alexandria, Egypt; ali.hamdy343@yahoo.com; 2Fabrication Technology Research Department, Advanced Technology and New Materials Research Institute (ATNMRI), City of Scientific Research and Technology Application (SRTA-City), New Borg El-Arab City P.O. Box 21934, Alexandria, Egypt; shafez@srtacity.sci.eg

**Keywords:** delignification, lignocellulose, ultrasonic irradiation, autoclaving, response surface methodology, enzymatic hydrolysis, corn stover, bioethanol

## Abstract

Bio-based treatment technologies are gaining great interest worldwide, and significant efforts are being afforded to develop technology for the use of lignocellulosic biomass. The potential of corn stover (CS) as a feedstock for bioethanol production was investigated by creating an optimal pretreatment condition to maximize glucose production. The current study undertook the impact of novel physico-chemical pretreatment methods of CS, i.e., autoclave-assisted oxalate (CSOA) and ultrasound-assisted oxalate (CSOU), on the chemical composition of CS and subsequent saccharification and fermentation for bioethanol production. The delignification was monitored by physicochemical characterizations such as SEM, XRD, FTIR, CHNs, and TGA. The results evidenced that delignification and enzymatic saccharification of the CS pretreated by CSOA was higher than CSOU. The optimum enzymatic saccharification operating conditions were 1:30 g solid substrate/mL sodium acetate buffer at 50 °C, shaking speed 100 rpm, and 0.4 g enzyme dosage. This condition was applied to produce glucose from CS, followed by bioethanol production by *S. cerevisiae* using an anaerobic fermentation process after 72 h. *S. cerevisiae* showed high conversion efficiency by producing a 360 mg/dL bioethanol yield, which is considered 94.11% of the theoretical ethanol yield. Furthermore, this research provides a potential path for waste material beneficiation, such as through utilizing CS.

## 1. Introduction

According to the UN sustainable development goals and the challenges of the energy shortage, the utilization of agricultural perspectives as renewable energy resources is a target for a sustainable economy [1]. For a more green economy, renewable biomass such as crop residues are increasingly being utilized to produce chemicals, fuels, and energy; this transition to a more sustainable economy is also a fundamental problem for society nowadays. Environmental issues such as price hikes of petrol-based resources and climatic change and the limited nature of oil supplies are the driving force that is causing the shift from a fossil-based to a bio-based economy. According to EU-27 Switzerland, by using the geographic estimation, we can classify the residual biomass into four categories: agricultural, forestry residues, municipal biodegradable wastes, and urban greenery management. As a result, natural resources, particularly agricultural resources, will play a significant role in producing biofuel and bioenergy from biomass, allowing fossil resources to be replaced while maintaining the merits of overall sustainability [2,3].

Lignocellulosic materials are one of the most promising alternative energy sources for bioethanol production. The primary source of lignocellulosic materials is in crops such as palm tree [4], sugarcane [5], wheat straw [6], rice husk [7], and corn stover [8]. All these lignocellulosic materials mainly contain three components: cellulose, hemicellulose, and lignin. The percentage of each element varies according to the plant type, species, and parts. Mainly, corn is abundantly cultivated in a vast amount worldwide (approximately 1.06 × 10^9^ tonnes/year) [9]. The corn crop is considered one of Egypt’s main strategic crops as its production reached 6.8 million tons in 2020. The latest reports and statistics of the Ministry of Agriculture and Land Reclamation reported that 1 acre produces 600 kg of corn stover, which is considered a waste. Hence, the farmers burn this massive waste, leading to severe environmental problems, global warming, and health deterioration. Producing valuable products such as bioethanol from these affordable wastes avoids enormous health and ecological harms, reduces the bioethanol production cost, and saves on waste transportation and disposal costs [10]. Pretreatment, hydrolysis, and fermentation are the three critical stages in bioethanol production from lignocellulose. Among these, pretreatment is the most challenging step for improving the accessibility to subsequent hydrolysis and fermentation [11,12].

The supramolecular polymer structure of these lignocellulosic components is highly obstinate to degradation. The pretreatment process is performed to disrupt the complex lignocelluloses and separate lignin and hemicellulose fractions from cellulose. The detrimental effects of lignin on the bioconversion process might manifest themselves in a variety of ways: (1) lignin might coat the surface of a reactive carbohydrate and/or form lignin–carbohydrate complexes (LCC), (2) nonproductive enzyme binding, and (3) soluble lignin fragments inhibiting a free enzyme in solution [13]. Moreover, the main barrier for crop bioethanol production is the recalcitrant nature of lignocellulosic materials, including lignin and hemicelluloses. Moreover, the high crystallinity index (CI) of cellulose bonds between cellulose bonds lignin and cellulose components also obstacles bioethanol production. For the reduction of the adverse effect of this barrier and produce cellulose-digestible substrates, pretreatreatment has been applied to the biomass [14,15].

Although the alkaline pretreatment utilizes lignocellulosic materials without adverse environmental impacts, it requires secondary acid treatment. Hence, acid treatment has been proven to be a hopeful process in the effective delignification process. The pretreatment using acids reduce the cellulose and hemicelluloses resistance to physical, chemical, and biological collapse. Although mineral acids show effective lignin removal, they have many drawbacks such as (1) a negative environmental effect, (2) the effluent acid being non-recoverable, (3) retardation of the fermentation process in some industries, and (4) high energy and chemical cost [16,17].

Carboxylic acid pretreatment is a promising technology that can dissolve the lignin in the pretreatment process without affecting cellulose. Carboxylic acid could be used as an alternative to the mineral acids as they have several advantages such as (i) the produced lignin being characterized by its high purity and recoverability, which allows it to be used in many applications; (ii) the produced lignin being sulfur-free and containing a narrow range of low molecule weights; and (iii) the process potentially being operated in small-scale plants with low-cost investment. For decades, several types of organic matter with different functional groups and polarities have been studied for their efficiency and selectivity in extracting lignin from various lignocellulosic materials. In addition, several strategies have considered the co-treatment, such as using chlorine chloride with a different carboxylic acid for pinewood sawdust delignification [18]. Other scientists used hydrogen peroxide sole and acetic acid for the corn stover delignification process [9].

The critical bottleneck in developing a highly efficient delignification protocol remains the development of the effective pre-treatment process. Although chemical techniques can obtain the pretreated biomass, much longer treatment times are required for the preferred conversions. As a result, a single pretreatment method is not always successful. Multiple pretreatment approaches have lately been recognized as potential options for increasing the delignification efficiency and bioethanol yields.

The processing of lignocellulosic biomass has been done with a variety of pretreatment systems. Ultrasound-assisted chemical reactions can be a brilliant methodology for intensifying chemical reactions (benefits such as slower reaction times and lower quantum of chemicals). The preference for ultrasound-induced cavitation results from removing the mass transfer resistance based on the acoustic sound turbulence streaming. Acoustic cavitation creates, expands, and implosively collapses microbubbles in the ultrasound-irradiated liquid. This feature is the most effective under high-power and low-frequency operation [19].

Another strategy for developing effective delignification is to use heat to facilitate the breakdown of lignin–cellulose bonds during chemical action. One of the most effective heating techniques for pretreatment is moist heat under pressure from an autoclave. Compared with other heating sources, the autoclave supplies both heat and stress as well. This feature successfully makes the lignocellulosic biomass recalcitrance structure vulnerable to chemical action. The air moisture helps evenly spread heat throughout the autoclave chamber, improving the heating process effectiveness [20].

Nevertheless, no study on the combination of autoclaving with a mild oxalic acid solution as a pretreatment method for bioethanol production has been found to the best of the authors’ knowledge. In the present work, the objectives were to study the benefits of combining oxalate with autoclaving over the ultrasound irradiation for establishing the intensification of the chemical delignification process, which forms the novelty of the current work. The efficacy of comparing ultrasonic irradiation and autoclaving for delignification in the presence of oxalic acid treatment was carried out in terms of physicochemical characteristics. Central composite rotatable design (CCRD)-based RSM was employed to optimize the process conditions for achieving maximum glucose production yield. This optimum condition was used for the fermentation process by *S**. cerevisiae* for total bioethanol production.

## 2. Materials and Methods

### 2.1. Sampling and Processing of CS Biomass

The CS used in this study was collected from the local farms in Beheira Governorate, Egypt. The CS biomass was chopped into pieces and then washed and dried at 60 °C overnight in an oven (Nabertherm GmbH, Lilienthal, Germany) for 72 h until the constant weight was obtained. The dried samples were grinded in Kleinfeld, Germany, and passed through <250 μm screen using a Retsch AS200 (Haan-Gruiten, Germany). The sieved biomass was stored in tightly closed plastic bags at room temperature until usage. The sample was labelled as CS.

### 2.2. Autoclave-Assisted Oxalate Pretreatment

Autoclave pretreatment was assisted by using oxalic acid as a catalyst. The dried CS powder (5 g) was suspended in 2% oxalic acid (1:20 w/v) inside tightly sealed autoclavable bottles to avoid moisture loss. The autoclave operated at 120 °C for 60 min using a Tuttnaur autoclave-steam sterilizer (3850 EL, Horsham, UK) under pressure of 310 kPa. At the end of the pretreatment process, the mixture was rapidly separated by vacuum filtration and washed with hot water several times until neutralization. Then, it was oven-dried at 50 °C for 12 h until a constant weight was reached. After this, it was kept in a desiccator for characterization and further processing. This sample was labeled as CSOA.

### 2.3. Ultrasonic-Assisted Oxalate Pretreatment

The same procedure mentioned in Section 2.2 was followed, except the mixture was located at an ultra-sonication water bath (J.P. Selecta, Bercelona, Spain) for 20 min, with temperature adjustment to 60 °C; this sample was labeled as CSOU.

### 2.4. Estimation of Lignin, Hemicellulose, and Cellulose

The CS and pretreated solid samples, CSOA and CSOU, were used for determining the solid recovery yield using Equation (1) [21].



(1)
$$\%\;Solid\;recovery\;yield=\frac{Final\;pretreated\;weight\;\left(g\right)}{Initial\;CS\;weight\;\left(g\right)}\;\times \;100$$



The chemical composition of the pretreated samples was determined by the Van Soest method [22]. About 1 g of dried treated residue was suspended in 70 mL neutral detergent and autoclaved at 100 °C for 40 min and 120 °C for 20 min. The resulting mixture was filtered, washed with hot water until it reached pH 6.5–7, washed with ethanol twice, dried at 50 °C overnight, and weighed; it was referred to as W_0_. The W_0_ sample was treated with 70 mL of 2 mol/L HCl and autoclaved at 100 °C for 60 min. The resulting mixture was filtered, washed with hot water until it reached pH 6.5–7, washed with ethanol, dried at 50 °C overnight, and weighed; it was referred to as W_1_. The W_1_ sample was soaked in 72% H_2_SO_4_ for 4 h at room temperature; then, water was added to the mixture and it was incubated overnight at room temperature. The mixture was filtered, washed with hot water until it reached pH 6.5–7, dried at 50 °C overnight, and weighed; it was referred to as W_2_. Finally, the W_2_ sample was burned in a muffle furnace (Barnstead Thermolyne 48000, Essex, UK) at 575 °C for 4 h. The resulted powder was weighed and was referred to as W_3_.

The content of hemicellulose, cellulose, lignin, and ash was estimated using the following Equations (2)–(5):(2)$$Hemicellulose=W_{0}-W_{1}$$
(3)$$Cellulose=W_{1}-W_{2}$$
(4)$$Lignin=W_{2}-W_{3}$$
(5)$$Ash=W_{3}$$

After determining the chemical composition, we calculated the % lignin removal, % hemicellulose removal, and % cellulose recovery according to Equations (6)–(8).(6)$$\%\;lignin\;removal=\left(\frac{L_{I}-\left(L_{F}\times \%\;\mathrm{Solid}\;\mathrm{recovery}\;\mathrm{yield}\;/\;100\right)}{L_{I}}\right)\times 100$$
where % lignin removal is the percentage of lignin removed after the pretreatment process compared with the initial lignin content, *L_I_* is the percentage of initial lignin content, and *L_F_* is the percentage of final lignin content in the solid residue.(7)$$\%\;Hemicellulose\;removal=\left(\frac{H_{I}-\left(H_{F}\times \%\;\mathrm{Solid}\;\mathrm{recovery}\;\mathrm{yield}\;/\;100\right)}{H_{I}}\right)\times 100$$
where % hemicellulose removal is the percentage of hemicellulose removed after the pretreatment process compared with the initial hemicellulose content, *H_I_* is the percentage of initial hemicellulose content, and *H_F_* is the percentage of final hemicellulose content in the solid residue.(8)$$\%\;Cellulose\;recovery=\left(\frac{C_{F}\times \%\;\mathrm{Solid}\;\mathrm{recovery}\;\mathrm{yield}\;/\;100}{C_{I}}\right)\times 100$$
where % cellulose recovery is the percentage of remaining cellulose after the pretreatment process compared with the initial cellulose content, *C_I_* is the percentage of initial cellulose content, and *C_F_* is the percentage of final cellulose content in the solid residue.

### 2.5. Characterization of Pretreated Samples

Fourier transform infrared (FTIR) spectroscopy was utilized to investigate the functional groups attached to the sample surface using a Shimadzu FTIR–8400 S (Kyoto, Japan). The detection was conducted in a wave range of 4000–450 cm^−1^, with 4 cm^−1^ resolution. The CHNS/O elemental analysis was carried out using a Vario-Micro CHN elemental analyzer (Elementar Analyses system GmbH, Langenselbold, Germany). X-ray diffraction (XRD) analysis was conducted using a powder X-ray diffractometer (Shimadzu-XRD-7000 Diffractometer, Kyoto, Japan). The crystallinity index (CrI) value by Segal method was provided as [(I_002_ − I_am_)/I_002_] × 100, where I_002_ symbolize peak with the maximum intensity of crystalline cellulose I at 2θ, ranging between 22 and 23°, with I_am_ corresponding to a peak with minimum intensity for crystalline cellulose I at 2θ, ranging between 18 and 19° [23,24]. A scanning electron microscope was used to obtain the surface morphology images (SEM, JEOL Model JSM 6360 LA, Tokyo, Japan). The thermal stability of samples was explored by thermogravimetric analysis (TGA) using a Shimadzu TGA-50 (Kyoto, Japan). The investigation was conducted in a nitrogen atmosphere from room temperature to 800 °C at a heating rate of 10 °C min^−1^.

### 2.6. Enzymatic Saccharification

Enzymatic saccharification was performed by mixing 1:25 g solid substrate/mL sodium acetate buffer (0.05 M, pH 5) in the presence of cellulase enzyme (Novozymes, Denmark) 20 FPU/g dry substrate. The enzymatic saccharification was optimized using different enzyme loading (0.1, 0.2, 0.3, 0.4, 0.5, 0.6, 0.7, 0.8, 0.9, and 1 g) cellulose enzyme/g substrate and different incubation times (0, 24, 48, 72, 96 h) at 50 °C and shaking speed 100 rpm in a shaker incubator (IKA KS 4000-I control, Staufen, Germany). The optimum enzymatic saccharification operating condition was utilized in all experiments. After the enzymatic saccharification step, the mixture was centrifuged, and 10 uL of supernatant was obtained and mixed with 1 mL of glucose buffer (glucose kit, Bio-Med). This mixture was incubated for 15 min at room temperature. The glucose production yield was estimated using spectrophotometric analysis (Spectrophotometer 7230 G, Shanghai, China) by measuring the absorbance at λ_max_ = 546 nm.

### 2.7. Modeling and Statistical Data Analysis

The experimental results of the glucose production yield for all experiments were presented as mean (M) ± standard deviation (SD). Calculations of M and SD were carried out using Microsoft Excel. The effect of process variables such as temperature, oxalic acid concentration, contact time, and the solid–liquid ratio on glucose production yield was investigated using the RSM model and CCRD design. These independent variables were selected at five levels, as shown in Table 1.

A fully functional CCRD model with five levels and three factors was established, and 26 experimental conditions were designed and randomized to reduce variability effect of the produced response using JMP version 40.4 Copyright © 1989–2001, SAS Institute Inc., Cary, North Carolina, USA, to reduce variability effect in the total reducing sugars and glucose yield as the responses. The empirical formulation of the second-order mathematical model to fit the process equation is presented in Equation (9). The resulted from experimental data were analyzed using the regression method (Microsoft Excel) to describe the quadratic model of the response. Analysis of variance (ANOVA) was applied to evaluate the features of the fitted model.
*Y* = Ω_o_ + Ω_1_A + Ω_2_B + Ω_3_C + Ω_4_D + Ω_12_AB + Ω_13_AC + Ω_14_AD + Ω_23_BC + Ω_24_BD + Ω_34_CD + Ω_11_A^2^ + Ω_22_B^2^ + Ω_33_C^2^ + Ω_44_D^2^(9)
where *Y* is the glucose production yield (response); Ω_o_ is the intercept term; Ω_1_, Ω_2_, Ω_3_, and Ω_4_ are the coefficients of the linear terms; Ω_12_, Ω_13_, Ω_14_, Ω_23_, and Ω_24_ are coefficients of the interaction terms; and Ω_11_, Ω_22_, Ω_33_, and Ω44 are coefficients of the quadratic terms. A B, C, and D are the independent variables. The experimental design results were investigated and interpreted by STATISTICA 7 (StatSoft, Dell, TX, USA) statistical software to estimate the response changes according to independent variables changes.

### 2.8. Fermentation Process

For inoculum preparation, Satccharomyces cervisiae was transferred from slant and cultivated in 100 mL of YPD medium at 30 °C and 150 rpm for 36 h [25].

The fermentation process was performed under anaerobic conditions using a layer of paraffin oil upon the fermented medium in 100 mL conical flasks containing 50 mL enzymatic hydrolysate supplemented with the fermentation medium components (KH_2_PO_4_ 1 g/L, (NH_4_)_2_SO_4_ 5 g/L, MgSO_4_ 0.5 g/L, and yeast extract 1 g/L). The prepared medium was sterilized at 121 °C for 20 min. Batch fermentation was initiated by inoculating 5% (v/v) inoculum seed (yeast) into 50 mL of the medium and covered by a paraffin oil layer in a 100 mL conical flask fitted with plugs covered by parafilm. The flasks were statically incubated at 30 °C for 72 h. After fermentation, the medium was centrifuged at 10,000 rpm for 10 min, and the supernatant was used to determine the bioethanol production yield using the chromic acid method [26]. The results were also emphasized using HPLC Agilent 1100 (Santa Clara, CA, USA). The carrier gas was nitrogen, with a flow of 0.2 mL/min. The used column was Hi-plexCa uspl_19_ (Agilent) 250 × 4 m and operated at 80 °C. The detector was RI and operated at 55 °C.

## 3. Results and Discussion

### 3.1. Combined Physical and Chemical Pretreatment

Even though CS has been considered an economical and available lignocellulosic biomass for producing high-value products, including bioethanol, it essentially needs pretreatment conditions to break down its structure and enhance cellulose accessibility for enzymatic saccharification and, consequently, high bioethanol production. Many other researchers have discovered that lignin is a significant inhibitor of enzymatic saccharification in CS instead of other herbaceous plants (such as rice straw), where hemicellulose is the predominant inhibitor.

An optimal pretreatment lowers the barriers that cause lignocellulosic recalcitrance and transforms the biomass into liable forms for enzymatic accessibility, resulting in high hemicellulosic sugar recovery with fewer inhibitors. Physical pretreatment generally disrupts the lignocellulosic structure, while chemical pretreatment separates and removes lignocellulosic components. As a result, the combination of the two different pretreatments improves the effectiveness of the enzymatic saccharification. The CS pretreatment was preliminarily carried out by oxalic acid in association with autoclaving or ultrasonic pretreatment. These pretreatments usually disrupt glycosidic linkage of/between hemicellulose and lignin, resulting in hemicellulose dissolution, which enhances the enzymatic accessibility of cellulose. In summary, the combined physical and chemical pretreatment breaks the complex structure of CS, thus changing the morphological features, surface area and porosity, crystallinity index, and degree of polymerization, which reduces the biomass recalcitrance.

### 3.2. Effect of Physicochemical Pretreatments on CS

#### 3.2.1. Composition of CS and Pretreated CS

After pretreatment, the lignocellulosic compositions of the treated samples are summarized in Figure 1 to be compared with CS. The combined physico-chemical pre-treatment increased the cellulose content from about 36% to 57% for CSOA and 40% for CSOU, while the lignin content decreased from 20% for CS to 12% for CSOU and 11% for CSOA. This shows that ultrasonic-assisted oxalate pretreatment removed 55% of lignin and 26% of hemicellulose and retained about 71% of the cellulose. Nevertheless, a significant fraction of the lignin and hemicellulose remained in the solid residue after this pretreatment. However, autoclave-assisted oxalate pretreatment presented higher delignification (67%) and higher hemicellulose dissolution (64%) while maintaining more significant amounts of cellulose (92%) in the residue. The high cellulose content in the CSOA sample indicates that the combination between the autoclave, as a kind of physical pretreatment, and oxalic acid, as a chemical pretreatment, selectively dissolve lignin and hemicellulose, as well as protect and fractionate the cellulose. However, the combination between the ultrasonic irradiation, as a kind of physical pretreatment, and oxalic acid, as a chemical pretreatment, led to lower cellulose recovery and dissolved lower lignin and hemicellulose. Hence, the cellulose content in the CSOU was lower than in CSOA.

#### 3.2.2. Fourier Transform Infrared Spectrum Analysis

The breaking of bonds and modification in the stretching vibrations related to the structural integrity of lignocellulosic biomass was presented by FTIR spectra. The FTIR spectra of CS before and after physico-chemical pretreatment indicated the structure changing due to the peak shape and intensity change, as shown in Figure 2. All the samples show a band around 3400 and 3300 cm^−1^ corresponding to O–H symmetric and asymmetric stretching vibration that indicates the presence of alcohols, phenols, and physisorbed and chemisorbed water molecules in cellulose [27,28]. After pretreatments, this peak became broader, indicating more vibrational modes of inter- and intra-molecular hydrogen bonding and lower amounts of OH groups [29]. The peaks at 2913 cm^−1^ are presented in the three samples and contribute to the aliphatic structures of asymmetrical C–H stretching vibrations in lignin and/or hemicelluloses [30,31]. The intensity of this peak in CS > CSOU > CSOA indicates a significant decrease in the lignin and/or hemicellulose content in the treated samples in comparison with the CS. The peak at 2349cm^−1^ that appeared in the CSOU sample corresponded with the –OH stretching vibration mode of the hydroxyl functional groups [32,33]. The peak at 1740 cm^−1^ in the CS sample was related to C=O stretching vibration of either the uronic and acetyl ester groups of the hemicelluloses or the ester linkage of the carboxylic group the ferulic and p-coumaric acids of lignin, pectin, and/or hemicellulose [34,35]. This peak completely vanished after the pretreatment process, indicating efficient removal of most of the lignin and hemicellulose from the CS. The peaks at 1628 cm^−1^ were related to the C = C aromatic skeletal stretching vibrations and C=O stretching of lignin [36]. The intensity of this peak decreased after the pretreatment process, which indicates that the lignin content was reduced due to a successful delignification process that resulted from C=O to C–OH group transformation. In the CS sample, the peak at 1234.9 cm^−1^ was related to guaiacyl/syringyl ring and C–O stretching vibration, which indicates the presence of both syringyl and guaiacyl in residual lignin and/or xylan [37]. This peak was shifted to 1230 cm^−1^ in CSOU sample, which could have been due to C = O absorption that resulted from cleavage of acetyl groups [38]. This peak wholly disappeared in the CSOA, which meant significant removal of lignin. The vibration of the sharp peak at 1031 cm^−1^ was related to C_1_-H deformation with ring vibration contribution, C–O and C–C stretching, and C–OH bending in β-(1-4)-glycosidic linkages between glucose and cellulose [29,39]. The intensity of this peak in the CSOA > CS > CSOU indicated a significant increase in the cellulose content after the pretreatment using an autoclave. However, using ultrasonic irradiation potentially led to breaking the cellulose and dissolving it, as occurred with lignin and hemicellulose. This means that pretreatment using autoclave decreased the lignin and hemicellulose effectively without deteriorating the β-(1-4)-glycosidic linkages between glucose and cellulose. Two additional peaks existed in the CSOA sample at 892 and 545 cm^−1^, attributed to cellulose and C–OH bending, respectively [40]. Therefore, FTIR analysis clearly confirmed the degradation of lignin and hemicellulose due to combined physical and chemical pretreatments.

#### 3.2.3. Elemental Characteristics

CHNS elemental analysis provided the elemental composition of the sample bulk structure for quantitative assessment of the compositional data of the samples and their degree of oxidation. The compositional data for CS, CSOU, and CSOA samples are listed in Table 2. Agricultural wastes, including CS, are mainly composed of cellulose, hemicellulose, and lignin. Thus, its primary element components are carbon, hydrogen, and oxygen. Although the CHNS analysis did not provide information about the O element, a rise in the weight percentage of C suggests a reduction in the O-content and, as a result, the pretreated sample’s oxidation state [41]. The O content was estimated from the mass difference.

After the pretreatment process, the carbon content in the CS < CSOU < CSOA may have been related to generating new hydroxyl groups in the sample, in addition to hydrolysis of methoxy groups and ether bonds. The nitrogen content in this instances was assigned to the remained protein from the cell wall. Hence, it is clear that the N content was reduced from 1.08 to about 0.45 after the pretreatment process, which indicates a weak chemical bond between proteins and the pretreated samples. The CSOA had the lowest atomic ratio of H/C, suggesting that this sample had the highest aromaticity with the highest aromatic structure [42]. Otherwise, increasing this ratio in the other samples indicates that they might contain a relatively higher content of initial organic components, such as lignin, fatty acids, and polymeric CH_2_ (aromatic core) [42,43]. The changes in H/C and O/C observed in the pretreated samples suggested that the structural characteristics were affected by the physico-chemical pretreatments and proved the high efficiency of the delignification process.

#### 3.2.4. Structural Characteristics

XRD provided more insights into the structural improvement after the pretreatment process, as well as the inter/intramolecular linkage alterations that resulted in cellulose swelling, lignin removal, and enhanced accessibility to enzymes. In Figure 3, the diffraction peak is approximately located at 2θ = 21.69° (002), indicating the existence of cellulose type II [29]. It can be seen that the CrI of the CS was determined to be 45.73%, which subsequently boosted to 56.59% after autoclave-assisted oxalate pretreatment. The increase in CrI was due to a reduction in amorphous fractions, primarily lignin and hemicellulose, which exposed the reactive cellulose. Contrarily, the CrI of CSOU was reduced from 45.73 to 39.5%, which indicates the cleavage of the crystalline structure of cellulose in CS as a result of the ultrasonic irradiation. It is claimed that the autoclaving pretreatment was influential in the depolymerization of amorphous lignin that resulted in a higher percentage of crystalline cellulose, thereby improving the rate of enzymatic saccharification [44]. Hence, it can be deduced that the autoclave-assisted oxalate pretreatment process markedly fractionates the crystallinity of the cellulose.

#### 3.2.5. Morphological Features

As a result of CS structural complexity, its recalcitrance is a significant limiting factor that inhibits enzymes from accessing its polysaccharide components. When the pores of the biomass are large enough to accept enzyme components, the digestibility of lignocellulosic materials can be greatly improved. The impact of the pretreatment on the morphological structure of CS is shown in Figure 4. The CS sample, which acted as a control, exhibited a compact, densely packed, highly ordered, and continuous smooth surface with a low degree of porosity, as shown in Figure 4a. It tends to be self-assembled with micron width and micrometer length fibers covering and protecting the lignin located in the outer surface of cellulose and hemicellulose. The strong inter-fibrillary attraction was caused by the hydrogen bonding between surface hydroxyl groups of cellulose during the CS drying. The morphological feature of CS may act as an obstacle to the enzymatic attack and adsorption onto the surface, resulting in low enzymatic saccharification. However, applying oxalic acid as a chemical pretreatment on CS in combination with the physical pretreatment increased the fiber roughness and opened up the well-shaped fibrils and the cellulose fibers with different degrees. As a result, the lignocellulosic structure was degraded by disrupting the connections that connect the lignin–polysaccharide matrix, producing structural alterations and increasing the biomass porosity and surface area [45].

Furthermore, the combination between oxalic acid and the physical pretreatment exhibited uneven surfaces with cavities and cracks, as well as ruptured fibrillary network, due to the successful removal of lignin and hemicellulose amorphous regions from the CS. Moreover, the lignin layer in the case of CSOA (Figure 4c) was disrupted and stripped off with longitudinal cavities more than in the case of CSOU (Figure 4b), leading to higher porosity and partial visibility of cellulose fibers. CS pretreated surface was more open and exposing of the reactive spots of cellulose in order to improve enzymatic accessibility. The drastic ruptures and high roughness of the CSOA surface indicates a significant delignification process [46].

The change in the physical technique exposed to the sample noticeably appeared on the dimensions of these formed cavities, wherein the smallest width appeared in the CSOA sample (13.6 μm) with the highest repetition units. In comparison, it was 31.38 and 38.25 μm when exposed to ultrasonic waves. These changes in the surface morphology encourage the adsorption and saturation of the enzyme on the cellulose fiber, thereby supporting the efficiency of the enzymatic hydrolysis process. On the basis of the SEM images, we found that the combined effect of autoclave and oxalic acid was more efficient in disrupting the lignocellulose cell wall, removing lignin, increasing the accessibility of cellulose, and improving the enzymatic saccharification as well. These images evidenced the results presented from the XRD analysis.

#### 3.2.6. Thermal Stability

To evaluate the effect of the autoclave or ultrasonic pretreatment with oxalic acid on the delignification of CS, we conducted thermal stability and decomposition of organic polymers by TGA analysis. Generally, it is clear that three stages of weight loss represent the sample decomposition, as observed in Figure 5. Overall, the CSOU was highly thermally stable in comparison with CS or CSOA. Hemicellulose degradation was in the first stage, while the decomposition of cellulose was in the second and the third stages [47]. In detail, the first decomposition between 25 and 120 °C provided almost the same weight loss (6–7%) for all samples. This stage corresponded to the drying period, mainly non-dissociative, physically adsorbed water molecules as well as surface hydrogen bonding by water [39,48]. The second stage is related to the destruction of the crystalline components and the polymer decomposition, which increases the amorphous structure and decreases the degree of polymerization. For the CSOA and CSOU, the weight loss in the range between 120 and 393 °C was about 73.81%, while the weight loss was 51.04% in the case of CS in the same range. This weight loss was mainly due to the tar formation, composed chiefly of L-glucose [49]. The significant drop in this stage for CSOA and CSOU was attributed to the liberation of volatile hydrocarbon from rapid thermal decomposition of cellulose, hemicellulose, and some parts of lignin converted to combustible gases such as CO, CO_2_, and C_x_H_y_. These combustible gases are produced from the volatilization of various oxygen functional groups from their structure with different thermal stabilities, and their scission occurs at different temperatures [47].

On the other hand, there is another stage from 351 to 554 °C for CS, representing the degradation of lignin not obtained in CSOA and CSOU, indicating successful delignification processes [50]. For treated CS, the final weight-loss stage has occurred at the temperature range (400–800 °C). In this region, the crystalline components have been completely destroyed. The cellulose has been decomposed into the monomer of D-glucopyranose, which could be decomposed into free radicals and converted into volatiles and tar [49]. Within this stage, the degraded volatile compounds are derived from phenolics, alcohols, and aldehydes. Overall, the CSOU sample had the most outstanding thermal stability and had the highest char yield of 18%, while CSOA had a char yield of around 2.66%. These results revealed that CSOU and CSOA achieved the delignification process with high quality.

### 3.3. Enzymatic Saccharification

Enzymatic saccharification yield is the pivotal criterion for assessing the efficiency of the pretreatment process, and the results are shown in Figure 6.

Physical pretreatment plays an essential role in biomass fractionation to facilitate the action of the chemical pretreatment and increase its efficiency. The influence of CS pretreatment, either using an autoclave or ultrasonic with oxalic acid, on the enzymatic saccharification was investigated by estimating the produced glucose yields as presented in Figure 6a. The enzymatic saccharification was carried out using 2% oxalic acid, 1:20 solid to liquid ratio, and 1 h dual pretreatment reactions utilizing autoclave or ultrasonic techniques. Although a complete understanding of the architecture and recalcitrance of the biomass cell wall is still lacking, it is widely accepted that cellulose in the plant cell wall is wrapped and crosslinked with other non-cellulosic chemicals to produce the compact chemical structure of cellulosic biomass. It can be found that the glucose content showed a positive indication for the enzymatic saccharification [51]. Hence, the results show that the enhanced enzymatic saccharification and the increased produced glucose yield of CSOA are higher than CSOU. The higher glucose yield of 669.9 mg/dL was obtained for CSOA, compared to glucose yield of 288.7 mg/dL for CSOU after 96 h and enzyme dosage of 0.4 g. One plausible explanation for this observation could be the combination of pressure and steam from autoclaves form intense heat to destroy the recalcitrant structure of the CS than ultrasonic pretreatment [45]. Another reason for the reduced accessibility of cellulose microfibrils after ultrasonic-assisted oxalate pretreatment could be the localized cell wall collapse caused by the migration and extrusion of the solubilized lignin, unlike significant de-lignification of the pretreated residue obtained with CS autoclave-assisted oxalate pretreated samples [52].

For studying the influence of the enzyme dosage on the glucose production yield, we conducted enzymatic saccharification experiments using the identical amounts of 1:25 g solid substrate/mL sodium acetate buffer at 50 °C and shaking speed 100 rpm with different enzyme dosages as shown in Figure 6b. The substrate used to perform these experiments was CS treated with oxalic acid 2% and autoclave at 120 °C for 1 h with 1:20 solid to liquid ratio. It is clear from the data that glucose production yield increased with the increase in enzyme dosage up to 0.4 g to reach 546.5 mg/dL, then decreased until 0.6 g. Increasing the enzyme dosage to 0.7 g increased the glucose yield again to 528 mg/dL. After that, they the enzyme dosage grew from 0.7 to 1 g and did not significantly increase the glucose production yield. Hence, it can be decided to use 0.4 g of the enzyme dosage for further experiments.

### 3.4. Optimization of Pretreatment Operating Conditions Using Response Surface Methodology

From the results mentioned above, we can conclude that the combined effect of the oxalic acid and the autoclave leads to more efficient de-lignification and better enzymatic saccharification enhancement than utilizing the oxalic acid and the ultrasonic. Response surface methodology (RSM), a multivariable statistical approach, has been utilized to optimize and analyze the combined influence of numerous factors/variables connected to the respective process involved in lignocellulosic biomass. The present study designed RSM experimental model for the CSOA pretreatment under different operating conditions to optimize the enzymatic saccharification process. The variables and the response of the designed experiments of the RSM model are listed in Table 3. It can be seen that significant glucose yields were produced in experiments 12 and 24, the temperature range between 130 and 140 °C, the oxalic acid concentration range of 3–5%, pretreatment time from 30 to 60 min, and solid to liquid ratio between 1:30 and 1:40. From the RSM model, some relations can be elicited to investigate the effect of the different parameters on the enzymatic saccharification process.

The pretreatment temperature is one of the most critical factors that directly influence the enzymatic saccharification efficiency. For the same operating conditions, raising the pretreatment temperature enhances the enzymatic saccharification. Raising the pretreatment temperature increases the glucose production yield, as shown in Figure 7, which compares every two experiments that had the same operating conditions.

The average glass transition temperature (T_g_) of lignin was determined to be about 91 °C in a study conducted by Stelte et al. (2011), where the T_g_ attributes to the visco-elastic characteristics of an amorphous polymer, which is also known as the lignin structure in the CSOA in this context. The rigid and glassy configuration of lignin would be transformed into a rubbery state at temperatures above T_g_ due to an increase in the mobility of the polymer backbone and a partial reduction in the storage modulus, allowing the lignin framework to be softened and solubilized under mild conditions. Higher reaction temperatures softened the lignin components while also lowering the viscosity, resulting in enhanced de-lignification of CSOA [53].

### 3.5. Statistical Analysis

Statistical analysis of the parameters affecting the pretreatment process was used to obtain a focused understanding of the effect of each variable. For this, the glucose production yield after 96 h of enzymatic saccharification was chosen, at which time the glucose yield generally reached a maximum (Figure 6b). Depending on the pretreatment operating conditions (Table 3), the experimental glucose yield after 96 h of enzymatic saccharification ranged from 453 to 750 mg/dL. The model empirical equation relating the glucose yield as the response to the independent variables is described in Equation (10).
*Y* = 1364.12 − 14.49*A* + 80.39*B* − 5.16*C* − 16.1*D* − 0.66*AB*+ 0.02*AC* + 0.17*AD* + 0.69*BC* + 0.05*CD* + 0.07*A*^2^ − 4.29*B*^2^ − 0.01*C*^2^ − 0.09*D*^2^(10)
where *A* is the reaction temperature (°C), *B* is the oxalic acid concentration (w/v), *C* reaction time (min), and *D* is the solid–liquid ratio.

The significance of the components was determined by using *p*-value, which is statistical terminology used to describe the model’s effectiveness. Variables with a *p*-value of less than 0.05 are considered significant, with a 95% confidence level for each model [54]. Thus, the result of the application of analysis of variance (ANOVA) in Table 4 was used to evaluate the effects of the quadratic terms, individual terms, and their interactions on the glucose yield, as well as to evaluate the fitness and significance of the quadratic regression model. The ANOVA statistical results of the estimated effects on the glucose yield, together with the coefficients of the regression model, are tabulated in Table 4. All the products significantly affect the glucose yield (response), with *p*-value < 0.05, except the interaction between oxalic acid concentration (*B*) and solid to liquid ratio (*D*), which has *p*-value of 0.55. The removal of this interaction enhanced the model, as the significance levels of the other effects that were already significant were increased, without apparent *R^2^* value reduction, which changed from 0.9876 to 0.9871 (excluding the non-significant interaction (BD)). The solid to liquid ratio and the interaction between temperature and solid to liquid ratio (AD), the interaction between the oxalic acid concentration and time (BC), and the interaction between the time and the solid to liquid ratio (CD) showed highly significant effects on the glucose yield (response) with *p*-value < 0.05, as shown in Table 4. The high coefficient of determination (*R^2^*) implies that the model fits the data well, as described by the regression model equation. The high F-value of 71.1 and the low *p*-value < 0.05 of the model indicate that the model was statistically significant at the 95% confidence level [55].

Among the four pretreatment conditions that provided the best enzymatic hydrolysis with the highest glucose yield (experiments 24, 17, 11, and 22), an essential feature of pretreatments was the negative effect of the reaction time, as observed by experiments 11 and 22, which have the same operating condition except for the reaction time. It was clear that increasing the reaction time from 30 to 90 min decreased the glucose yield from 649 to 639 mg/dL. Hence, the pretreatment operation for 30 min would be optimum and would reduce the time and energy expenditure when applying 130 °C, 2% oxalic acid, and 1:40 solid to liquid ratio. The results also show that the highest glucose yield was produced by raising the pretreatment temperature to the highest value (140 °C), which created a glucose yield of 750 mg/dL. The positive effect of the oxalic acid concentration was significantly observed in experiments 22 and 17, which had the same operating condition except for the oxalic acid concentration. It was evident that increasing the oxalic acid concentration from 2 to 5% increased the glucose yield from 639 to 728 mg/dL. These results were proven by the statistical analysis, which confirmed the positive effect of oxalic acid concentration and the negative effect of reaction time on the response.

The correspondence plot between the actual and predicted glucose yield is depicted in Figure 8. The closeness of data points to the regression line indicates a good agreement between the actual glucose yield and the predicted values. Furthermore, the results signify a reasonable estimation of the empirical equation and the produced glucose yield (response) concerning changes of the independent variables (i.e., reaction temperature, oxalic acid concentration, reaction time, and solid–liquid ratio).

### 3.6. Optimization of the Glucose Yield

3D surface plots were utilized to represent the interactions of the four independent variables (temperature, oxalic acid concentration, reaction time, and solid to liquid ratio) on the glucose yield and concurrent consequences. Figure 9a shows that as both variables (temperature, time, and oxalic acid concentration) increased, the glucose yield increased until it reaches the maximum (≈750 mg/dL). Figure 9b–d shows that the highest glucose yield was produced at the reaction temperature from 130 to 140 °C. Moreover, decreasing the solid to liquid ratio to 1:10 increased the glucose yield, as shown in Figure 9b, which demonstrated that adequate liquid was necessary for lignin and hemicellulose dissolution. However, there was no significant increase in the glucose yield by decreasing the solid to liquid ratio from 1:20 to 1:40 using the same operating conditions in experiments 16 and 11, respectively. Hence, it can be considered that solid to liquid ratio in the range of 1:20–1:30, temperature of 130–140 °C, reaction time of 30–60 min, and oxalic acid concentration of 2–3% is the optimum range of operation to produce high glucose yield, as evidenced by experiments 11, 16, and 24. However, from an economic point of view, experiment 17 was excluded from the optimum operating range due to severe operating conditions (5% oxalic acid concentration, 90 min reaction time, and 1:40 solid to liquid ratio).

In this study, the excel solver was used to estimate the maximum glucose yield and the operating condition that led to it within the lower and upper limits of the ranges of the independent variables investigated in this study, i.e., the method did not search outside the ranges chosen. The excel solver suggested that the condition of 140 °C reaction temperature, 7% oxalic acid concentration, 120 min reaction time, and 1:50 solid to liquid ratio can produce the maximum glucose yield of 937.5 mg/dL. However, this condition was not economical because it used more than double the oxalic acid concentration, double-time, and about half the solid to liquid ratio of experiment 24 that produced a glucose yield of 750 mg/dL, i.e., severe operating conditions would be used with no substantial increase in the glucose yield. Hence, from an economic point of view, the working conditions of experiment 24 can be considered as the optimum operating condition.

### 3.7. Fermentation Process into Bioethanol

About 50 g of CS treated using the optimum operating condition of experiment 24 was hydrolyzed by cellulase enzyme using the optimum saccharification hydrolysis working condition: 1:30 g solid substrate/mL sodium acetate buffer at 50 °C, shaking speed 100 rpm, and 0.4 g enzyme dosage. Using the anaerobic fermentation process, we utilized the produced hydrolysate to determine the feasibility of bioethanol production by *S. cerevisiae*.

To investigate the conversion efficiency of the hydrolysate to bioethanol, we used pure culture as inoculum and glucose from the hydrolysate as the substrate for *S. cerevisiae* in the batch system using sterile flasks. The initial concentration of glucose was 750 mg/dL. After the fermentation process using *S. cerevisiae* for 72 h, the residual glucose decreased to 0.8 g/L, indicating high yeast conversion efficiency. The final ethanol yield from the CS hydrolysate fermentation was 360 mg/dL, which was considered 94.11% of the theoretical ethanol yield.

### 3.8. Overall Mass Balance

On the basis of CS composition analysis, we performed an overall mass balance diagram for each process, including pretreatment, enzymatic saccharification, fermentation, and bioethanol production (Figure 10). Accordingly, physico-chemical pretreatment of one kilogram of CS, containing approximately 357 g cellulose, 215 g hemicellulose, and 201 g lignin, yielded 579 g dry mass containing 327.7 g recovered cellulose, 73.5 g hemicellulose, and 63.11 g lignin, which indicated efficient hemicellulose and lignin removal. During enzymatic saccharification, approximately 231.6 g of cellulase enzyme was added to yield 750 mg/dL glucose. An anaerobic fermentation process was performed to convert the produced glucose to 370 mg/dL ethanol using *S. cerevisiae* cells.

## 4. Conclusions

This work demonstrates the first report on bioethanol production from corn stover by autoclave-assisted oxalate pretreatment to the best of our knowledge. This approach was proven to be a powerful technology for effective delignification at the operating conditions of 3% oxalic acid, 1:30 solid to liquid ratio, and 140 °C for 60 min. Physico-chemical characterization techniques such as FTIR, XRD, and TGA highlighted the functional groups broken during the delignification, proving beyond doubt the impact of autoclave-assisted oxalate pretreatment, thus emphasizing the decrease of CS lignin content. The effect of temperature, oxalic acid concentration, time, and solid to liquid ratio on glucose yield was investigated. The process optimization was achieved using RSM and CCD models. The pretreated substrates were subjected to enzymatic saccharification and showed more considerable enhancement for CSOA than CSOU, owing to the significant increase in cellulose accessibility, successful de-lignification, and lignocellulose porosity. Interestingly, 360 mg/dL of ethanol was produced as the best value, 94.11% of the theoretical ethanol yield. Therefore, this study provided a powerful strategy for the appropriate physico-chemical pretreatment and value-added cellulosic ethanol co-production in CS and beyond.

## Figures and Tables

**Figure 1 polymers-13-03762-f001:**
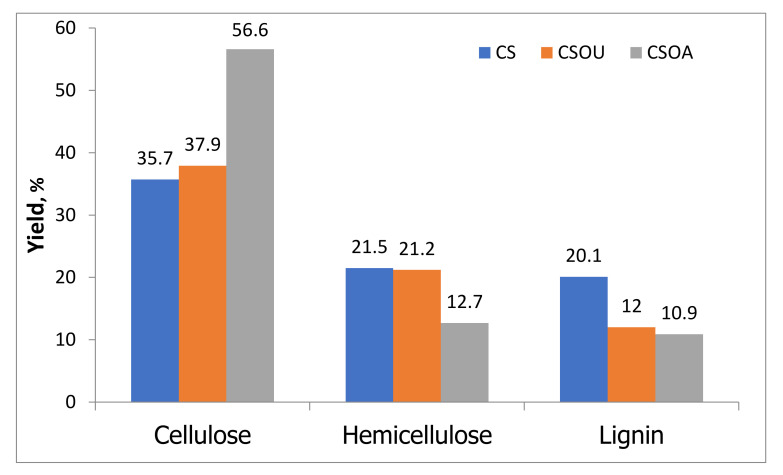
Lignocellulosic composition of CS, CSOA, and CSOU. The values are based on the total weight of untreated CS. Pretreatment condition: autoclave 120 °C, 2% oxalic acid, 1 h, and ultrasonic 60 °C, 2% oxalic acid, 1 h.

**Figure 2 polymers-13-03762-f002:**
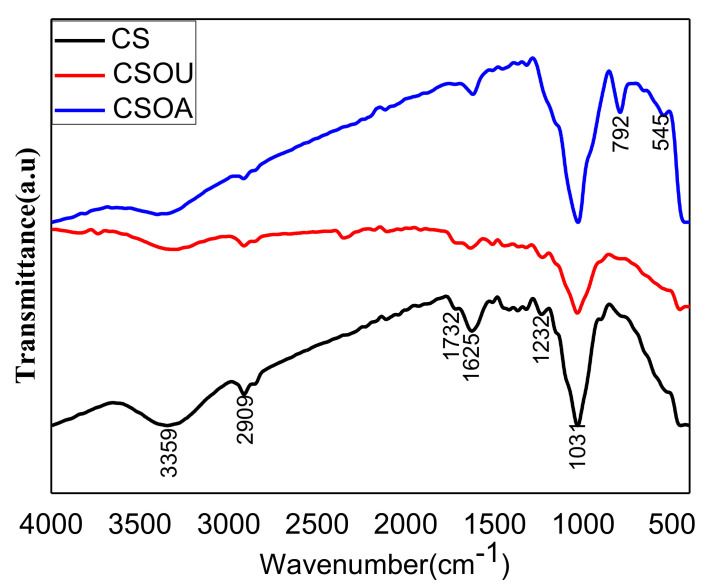
FTIR spectra of CS before and after physico-chemical pretreatments.

**Figure 3 polymers-13-03762-f003:**
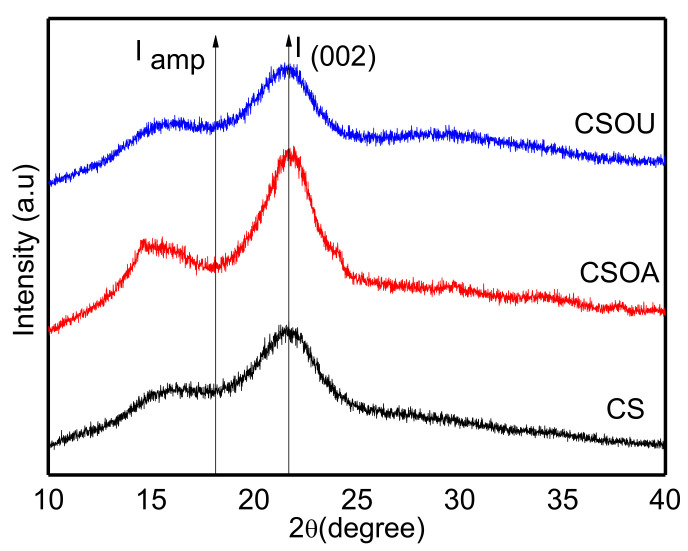
XRD patterns of the CS before and after various pretreatments.

**Figure 4 polymers-13-03762-f004:**
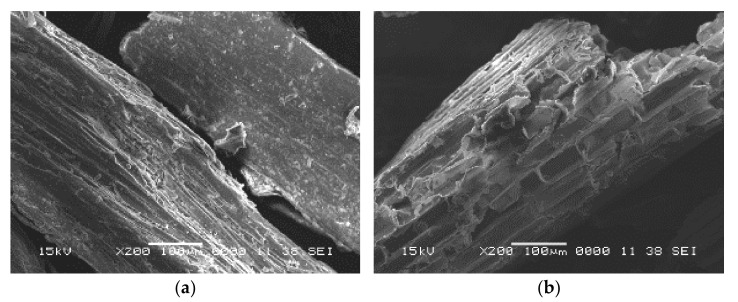
SEM images of (**a**) CS, (**b**) CSOU, and (**c**) CSOA.

**Figure 5 polymers-13-03762-f005:**
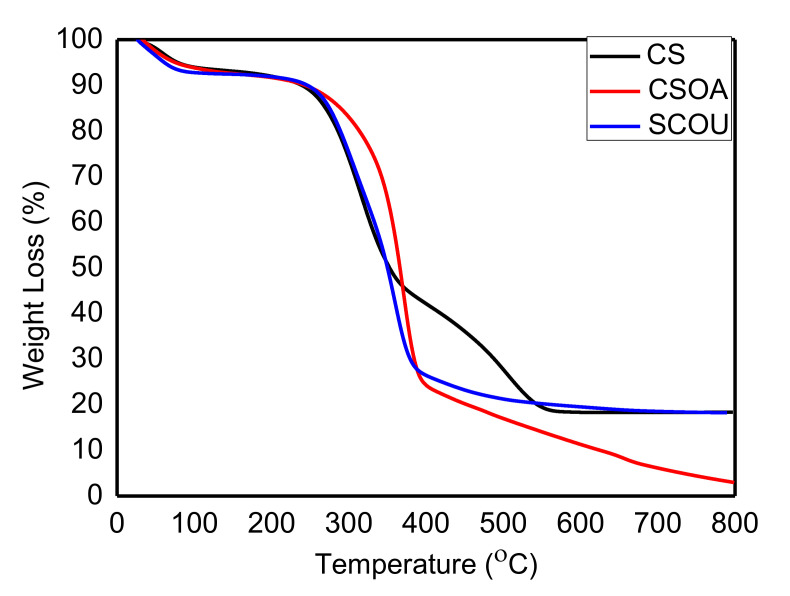
TGA of CS before and after physico-chemical pretreatments.

**Figure 6 polymers-13-03762-f006:**
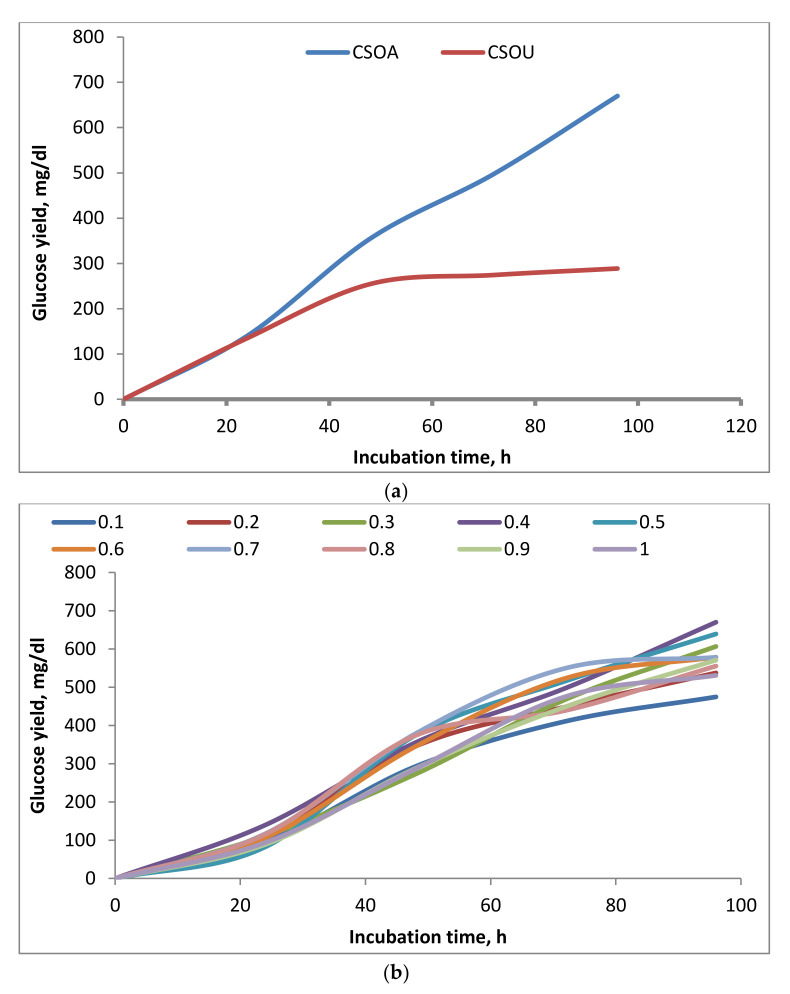
(**a**) Effect of physical pretreatment on the enzymatic saccharification, and (**b**) effect of enzyme dosage on the enzymatic saccharification enhancement.

**Figure 7 polymers-13-03762-f007:**
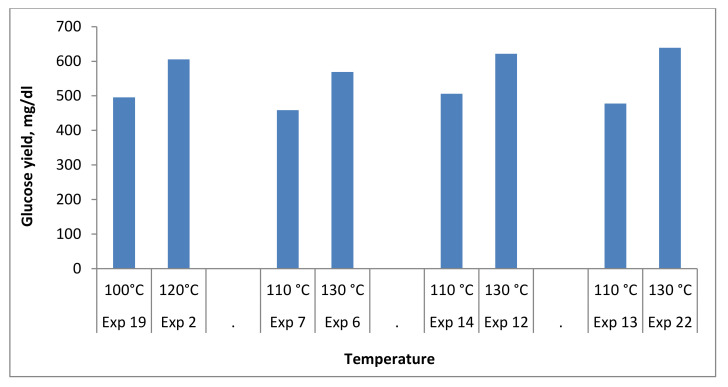
Effect of temperature on the glucose production yield.

**Figure 8 polymers-13-03762-f008:**
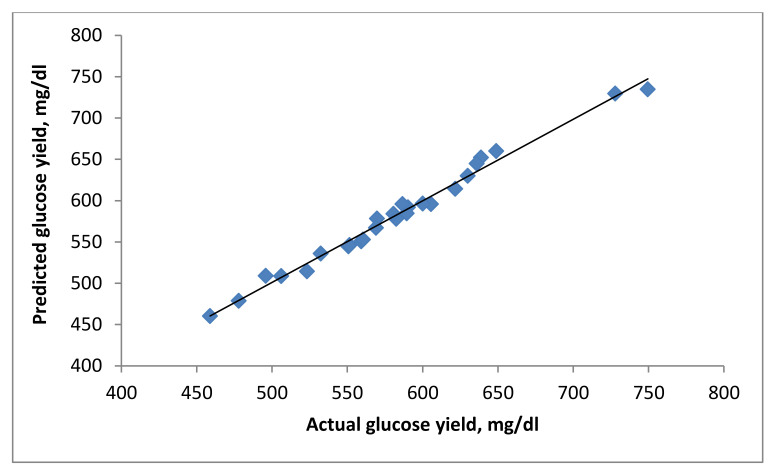
Regression plot between the actual and predicted glucose yield.

**Figure 9 polymers-13-03762-f009:**
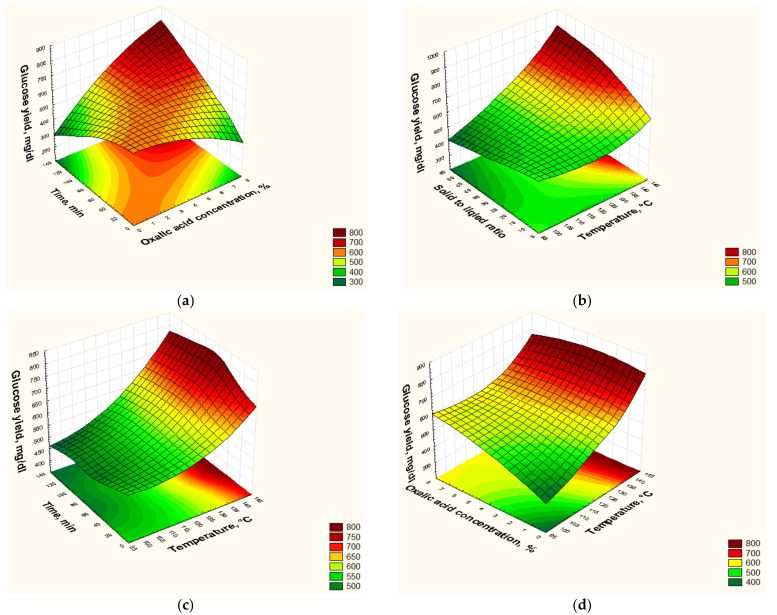
Interpretation of the effects of some variables by response surface of the interaction of (**a**) oxalic acid concentration with time, (**b**) temperature with solid to liquid ratio, (**c**) temperature with oxalic acid concentration, and (**d**) temperature with time.

**Figure 10 polymers-13-03762-f010:**
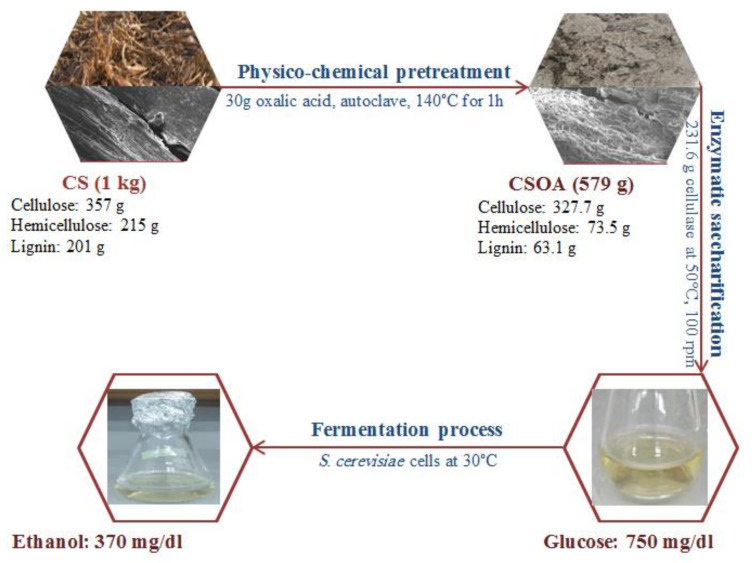
Mass balance for the production of bioethanol from corn stover.

**Table 1 polymers-13-03762-t001:** Levels of the independent variables.

Independent Variables	Levels				
	−2	−1	0	1	2
Temperature, °C (A)	100	110	120	130	140
Oxalic acid concentration, (w/v) (B)	1	2	3	5	7
Time, min (C)	10	30	60	90	120
Solid–liquid ratio (D)	1:10	1:20	1:30	1:40	1:50

**Table 2 polymers-13-03762-t002:** Elemental analysis of the untreated and treated samples.

Sample	N%	C%	H%	O%	H/C	O/C
CS	1.08	41.63	5.747	51.543	0.138	1.24
CSOU	0.44	42.75	5.827	50.983	0.136	1.19
CSOA	0.46	43.65	5.852	50.038	0.134	1.15

**Table 3 polymers-13-03762-t003:** Experimental design matrix of autoclave-assisted oxalate pretreatment, applied to CS, considering the variables temperature, oxalic acid concentration, time, and solid to liquid ratio, and the response, namely, produced glucose yield in 96 h of enzymatic saccharification.

No.	Temp, °C	Acid Concentration, %	Time, Min	Solid to Liquid Ratio	Experimental Glucose Yield mg/dL	Predicted Glucose Yield mg/dL
1	110	2	30	20	559.2	551.0
2	120	3	60	30	605. 6	595.8
3	110	2	30	40	523.3	514.7
4	120	7	60	30	590.4	592.1
5	120	3	10	30	569.7	579.0
6	130	2	90	20	569.0	567.2
7	110	2	90	20	458.9	460.5
8	110	5	90	20	582.6	578.0
9	120	3	120	30	560.4	553.0
10	120	3	60	30	589.6	595.8
11	130	2	30	40	648.8	660.2
12	130	5	30	40	621.7	614.4
13	110	2	90	40	477.9	479.0
14	110	5	30	40	506.2	508.7
15	110	5	30	20	550.7	544.9
16	130	2	30	20	630.0	630.1
17	130	5	90	40	727.8	729.7
18	120	3	60	50	589.5	584.6
19	100	3	60	30	495.9	509.3
20	130	5	30	20	580.7	584.3
21	110	5	90	40	600.0	596.4
22	130	2	90	40	638.8	652.1
23	130	5	90	20	635.9	644.9
24	140	3	60	30	749.5	734.9
25	120	1	60	30	551.4	546.2
26	120	3	60	10	532.4	536.1

**Table 4 polymers-13-03762-t004:** Estimated effects on the glucose yield, and regression coefficients of the model.

	Coefficients	Standard Error	*t*-Stat	*p*-Value
Intercept	1364.124191	414.9045024	3.2878	0.00648567
(A) Temp, °C	−14.48632433	6.305360802	−2.2975	0.04038268
(B) Acid concentration	80.38672663	23.61128492	3.40459	0.00522522
(C) Time, min	−5.163789976	1.22583554	−4.2125	0.00120492
(D) Solid to liquid ratio	−16.12015954	3.691088018	−4.3673	0.0009165
A^2^	0.065668409	0.025808449	2.54445	0.02572676
B^2^	−4.285919799	1.154182777	−3.7134	0.00296246
C^2^	−0.009623862	0.00328985	−2.9253	0.01271406
D^2^	-0.088706591	0.025808449	−3.4371	0.00492061
AB	−0.663406339	0.179896327	−3.6877	0.0031047
AC	0.022972021	0.009159873	2.5079	0.0275138
AD	0.166002438	0.027479618	6.04093	5.83899×10^−5^
BC	0.685950396	0.060137908	11.4063	8.49391×10^−8^
CD	0.045617062	0.009159873	4.9801	0.0003198

## Data Availability

The data presented in this study are available on request from the corresponding author.

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
