# Peer review of "The Interplay of Autoclaving with Oxalate as Pretreatment Technique in the View of Bioethanol Production Based on Corn Stover"

_polymers, 2021, doi:10.3390/polym13213762_

Round 1

Reviewer 1 Report

  1. Some language correction required; there are some grammatical errors and formatting problems. Please have another go of proof-reading. I think you show too many decimals in many places in the manuscript, e.g., in the compositional analysis. I do not believe that the accuracy is such that you can show 2 decimals. Please consider changing.
  2. line 247-248: As a crop, corn stover is not a high lignified biomass compared to softwood and hardwood.
  3. line 272-276: Please explain why the concent of glucan in the oxalate-assisted autoclave pretreated CS is higher than that in oxalate-assisted ultrasonic pretreatment.
  4. line 342-347: The CSOA has the lowest atomic ratio of  H/C, but it is almost the same as others. So it can not proved the high efficiency of the delignification .
  5. line 356-357: Please add the CrI of the CSOU and explain the difference.  
  6. line 381-382: Oxalic acid degrades the lignocellulosic structure by disrupting the connections between lignin-carbohydrate but the data shows low delignification between substrates.While there are no data illustrates that Oxalic acid degrades destroyed specifically.
  7. line 381-382: Oxalic acid degrades the lignocellulosic structure by disrupting the connections between lignin-carbohydrate but the data shows low delignification between substrates.While there are no data illustrates that Oxalic acid degrades destroyed specifically.
  8. line 426-427: If the degradation of lignin during 351 to 554 oC has been reported. If so, please cite it.
  9. In the whole manuscript, more latest references should be updated. You can check the following references

1). Unlocking the secret of lignin-enzyme interactions: Recent advances in developing state-of-the-art analytical techniques.

2). Understanding the effects of different residual lignin fractions in acid-pretreated bamboo residues on its enzymatic digestibility.

3). Revealing the influence of metallic chlorides pretreatment on chemical structures of lignin and enzymatic hydrolysis of waste wheat straw.

Author Response

Dear respected professor,

Sincerest thanks for your comments on our manuscript No.  polymers-1410563. We are very excited to have been given the opportunity to revise our manuscript. We carefully considered your comments. We want to extend our appreciation for taking the time and effort necessary to provide such insightful guidance. Based on your guidance, we have accordingly modified the manuscript and detailed corrections, changes and/or rebuttals against each point raised are listed below. Herein, we explain how we revised the paper based on your comments, also we offer detailed responses point by point.

Point 1: Some language correction required; there are some grammatical errors and formatting problems. Please have another go of proofreading. I think you show too many decimals in many places in the manuscript, e.g., in the compositional analysis. I do not believe that the accuracy is such that you can show 2 decimals. Please consider changing.

Response 1: Thank you for your valuable comments. The grammar and the formatting are checked for the whole manuscript. For the decimals point, the experiments in the manuscript were performed in duplicate or triplicate, so the final result is the average of these results which in many times result in decimals. But your comment has been considered and all the decimals are approximated. 

Point 2: line 247-248: As a crop, corn stover is not a high lignified biomass compared to softwood and hardwood.

Response 2: your esteemed note is right and this expression is removed.

Point 3: line 272-276: Please explain why the content of glucan in the oxalate-assisted autoclave pretreated CS is higher than that in oxalate-assisted ultrasonic pretreatment.

Response 3: The cellulose content increased in the CSOA than CSOU samples as a result of the lignin and hemicellulose dissolution and cellulose protection using the oxalic acid with autoclave than the oxalic acid with ultrasonic.  

“The high cellulose content in the CSOA sample indicates that the combination between the autoclave, as a kind of the physical pretreatments, and oxalic acid, as a chemical pretreatment, selectively dissolve lignin and hemicellulose, protect and fractionate the cellulose. However, the combination between the ultrasonic, as a kind of the physical pretreatments, and oxalic acid, as a chemical pretreatment, leads to lower lignin and hemicellulose dissolving and lower cellulose recovery”. 

Point 4: line 342-347: The CSOA has the lowest atomic ratio of H/C, but it is almost the same as others. So it can not prove the high efficiency of the delignification.

Response 4: In another research, “Chemical Pretreatment of Rice Straw Biochar: Effect on Biochar Properties and Hexavalent Chromium Adsorption” International Journal of Environmental Research (2019) 13:91–105, treatments lead to an atomic ratio of H/C differences about 0.02 and 0.03 and they considered those as differences to be mentioned to support their work.

In our work, this part (elemental characterization) is one of six characterizations proving the successful delignification. Hence, this work did not depend on this characterization alone, but we supported our results by it. 

Point 5: line 356-357: Please add the CrI of the CSOU and explain the difference.

Response 5:  Thank you for your valuable comments. The CrI of the CSOU is expressed in the text and the difference of CrI is also discussed in detail.

Point 6: line 381-382: Oxalic acid degrades the lignocellulosic structure by disrupting the connections between lignin-carbohydrate but the data shows low delignification between substrates. While there are no data that illustrates that Oxalic acid degrades destroyed specifically.

Response 6: Thank you for your valuable notification. This point is addressed in the section “3.2.5. Morphological features”.

Point 7: line 381-382: Oxalic acid degrades the lignocellulosic structure by disrupting the connections between lignin-carbohydrate but the data shows low delignification between substrates. While there are no data that illustrates that Oxalic acid degrades destroyed specifically.

Response 7: is the same to point 6

Point 8: line 426-427: If the degradation of lignin during 351 to 554 oC has been reported. If so, please cite it.

Response 8: The reference “Zdenka Kwoczynski *, Jirí Cmelík, Characterization of biomass wastes and its possibility of agriculture utilization due to biochar production by torrefaction process, Journal of Cleaner Production 280 (2021) 124302, is added.

Point 9: In the whole manuscript, more latest references should be updated. You can check the following references

1). Unlocking the secret of lignin-enzyme interactions: Recent advances in developing state-of-the-art analytical techniques.

2). Understanding the effects of different residual lignin fractions in acid-pretreated bamboo residues on its enzymatic digestibility.

3). Revealing the influence of metallic chlorides pretreatment on chemical structures of lignin and enzymatic hydrolysis of waste wheat straw.

Response 9:  Thanks a lot for valuable comments. All these references are stated in the manuscript as the reviewer suggested.

Finally, the authors want to thank reviewers for their valuable comments which are improved the scientific value of the manuscript

Best Regards

Hesham Hamad                                                                      

Reviewer 2 Report

This manuscript studies the combination of autoclaving with an oxalic acid solution as a pretreatment method for bioethanol production. The approach is novel and the study is thorough overall.

References are needed for "In addition, several strategies 
considered the co-treatment, such as using chlorine chloride with a different carboxylic acid for the Pinewood sawdust delignification" (line 88-89)

Author Response

Dear respected professor,

Sincerest thanks for your comments on our manuscript No.  polymers-1410563. We are very excited to have been given the opportunity to revise our manuscript. We carefully considered your comments. We want to extend our appreciation for taking the time and effort necessary to provide such insightful guidance. Based on your guidance, we have accordingly modified the manuscript and detailed corrections, changes and/or rebuttals against each point raised are listed below. Herein, we explain how we revised the paper based on your comments, also we offer detailed responses point by point.

 Point 1: References are needed for "In addition, several strategies considered the co-treatment, such as using chlorine chloride with a different carboxylic acid for the Pinewood sawdust delignification" (line 88-89)

Response 1: Thank you for your valuable comment. The authors added the reference (L. Li, L. Yu, Z. Wu, and Y. Hu, “Delignification of poplar wood with lactic acid-based deep eutectic solvents,” Wood Res., vol. 64, no. 3, pp. 499–514, 2019).

Finally, the authors want to thank reviewers for their valuable comments which are improved the scientific value of the manuscript

Best Regards

Hesham Hamad